# A Multivariate Analysis of the Interest in Starting Family Businesses within a Developing Economy

**DOI:** 10.3390/bs12060181

**Published:** 2022-06-07

**Authors:** Silvia Puiu, Roxana Maria Bădîrcea, Alina Georgiana Manta, Nicoleta Mihaela Doran, Georgeta-Madalina Meghisan-Toma, Flaviu Meghisan

**Affiliations:** 1Faculty of Economics and Business Administration, University of Craiova, 200585 Craiova, Romania; roxana.badircea@edu.ucv.ro (R.M.B.); alina.manta@edu.ucv.ro (A.G.M.); nicoleta.florea@edu.ucv.ro (N.M.D.); 2Faculty of Business Administration in Foreign Languages, Bucharest University of Economic Studies, 010731 Bucharest, Romania; madalina.meghisan@fabiz.ase.ro; 3National Institute of Economic Research “Costin C. Kiritescu”, Romanian Academy, 050711 Bucharest, Romania; 4Faculty of Economics and Law, University of Pitesti, 110014 Pitesti, Romania; flaviumeghisan2017@gmail.com

**Keywords:** family business, intergenerational, entrepreneurship, first generation, second generation, developing economy

## Abstract

The main objective of the research is to analyze the factors which influence the intention to start an intergenerational family business in a developing economy, highlighting the measures that can be implemented by decision-makers to stimulate these initiatives. PLS-SEM was used to analyze the data issued from 200 valid questionnaires. The survey was applied to 950 individuals from Romania. We focused on four variables: the closeness to family members, the financial support expected from family, the independence of individuals regarding the intention to start their own business and the intention to form partnerships with family members. All the hypotheses were validated, according to the final results. Thus, closeness to family members has a direct and positive impact on both the financial support expected from the family and the intention to form intergenerational family businesses. There is also a direct correlation between the financial support received from family and the intention to have partnerships with family members. Individuals who are closer to their families are not interested in developing independent businesses. There are several studies on family businesses in Romania, but there is no research analyzing the impact of closeness to family on the intention to develop an intergenerational family business. The study is useful for the decision-makers who can create national strategies in order to stimulate families to develop their own businesses.

## 1. Introduction

It is well known that entrepreneurship can contribute to economic development [1,2,3,4]. It can also solve social problems [5], allow technological advancement [6] and increase competitiveness and innovation [7]. According to European Union representatives, choosing an entrepreneurial career is a way to increase the integration of young people into the labor market and reduce the risk of social exclusion. Although, nowadays, young people are more interested in becoming entrepreneurs than the previous generations were, this is not a reference for starting a new business. Thus, EU-wide statistics show a higher interest in Romania in creating businesses compared to other countries, given that 29% of adults intend to start a business in the next three years, compared to the European average of only 13.56% [8]. This interest does not translate into a guarantee for the creation of new businesses, nor for their success over a longer period of time, since in Romania about a third of start-ups fail to reach the second year of activity [9]. 

In both developed and developing countries, family businesses represent the main form of entrepreneurship, with a major contribution to growth and job creation [10,11]. In fact, family businesses represent over 60% of European companies [12]. Family businesses have responded very well to the COVID-19 crisis; they have managed to persevere, representing a key factor for economic recovery [13]. In Romania, the emergence of family businesses is quite recent. Most emerged after 1990, and in the 31 years since then, the initially founded businesses have faced the need to be taken over by subsequent generations, making it difficult to determine the extent to which these businesses will continue to operate. In a study conducted by PwC Romania [14], 34% of the family businesses that were interviewed, considered succession planning to be a challenge for the organization and, in the case of 74% of family businesses, members of the younger generation already worked at the company. Thus, we also focus on the intention to achieve intergenerational family businesses in Romania and on the factors that influence these intentions, in order to highlight the measures that could be taken to stimulate these initiatives. 

## 2. Literature Review and Hypotheses Development

The present research analyses the interest in developing intergenerational family businesses within a developing economy, focusing on the relationship between four variables in this field: the closeness to family members; the financial support expected from family; the individuals’ independence regarding the desire to start a business on their own; the intention to form business partnerships with family members. Other studies have also analyzed the factors that impact the development of family businesses in Central and Eastern Europe [15,16]. Thus, Demirova and Ahmedova [15] (p. 18) highlight the role played by “shared values and beliefs”, “shared goals”, “family spirit”, financial independence, “flexibility” and specific organizational culture for the success of Bulgarian family businesses. Marjański and Sułkowski [16] conducted research on Polish family businesses emphasizing their importance for the country’s economy and also the fact that these small entities are sometimes better at satisfying the needs of niche markets than larger companies. 

The analysis of family businesses in Central and Eastern European countries should be seen through the lens of family relations and their role in these countries which have similar pasts marked by communism. Robila [17] mentions the importance of family ties in countries like Romania and Hungary where youngsters decide to remain in the same household with their parents because of their limited finances. This is an important motivation for uniting efforts to develop a family business [16]. Additionally, Robila [17] explains the strength of family relations in former communist countries like Romania because of experiences during the communist era in which you could trust only your close family. 

### 2.1. Conceptual Background

#### 2.1.1. The Closeness to Family Members

Several authors have analyzed the way that family support influences university students in choosing a business career and they have generally highlighted the positive relationship between the family environment and the desire to become an entrepreneur [18,19,20,21,22]. 

Scott and Twomey [18] analyzed options for university students. Their results identified the importance of parental influence and the way that students’ work experiences contribute to their decisions to pursue careers in entrepreneurship. Similarly, Peterman and Kennedy [23] have shown that students with previous family experiences in entrepreneurship show a greater interest in starting a new business. The influence of the family on the entrepreneurial personality was also studied by Maphosa [24], who observed that if parents encouraged behaviors that increased confidence, this could lead to risk-taking behavior and, thus, to the adoption of innovation, determining the development of an entrepreneurial personality in children.

Zellweger, Sieger and Halter [21] studied the importance of entrepreneurship education, underlying that it may be less effective in determining the entrepreneurial intentions of students who grew up with an entrepreneurial family, than with students without entrepreneurial families. This situation can be explained by the fact that students within an entrepreneurial family already have some experience and knowledge of the difficulties of being an entrepreneur. In addition, Huber [25] showed that different family relationships (parents, children and siblings) are particularly strong forms of social relationships that can influence the orientation towards entrepreneurship.

Studying the effects that family relationships have on the performance of family businesses, Adjei et al. [26] observed that the entrepreneur-child relationship is the dominant relationship in family businesses and family relationship affects the company’s performance, depending on the type of family connection. Laspita et al. [22] extended the analysis of the influence exerted by the family, from parents to grandparents and concluded that the relationship between parents and grandparents who have entrepreneurial experience generates models for grandchildren that will determine a strong preference for entrepreneurship. 

There are also other research papers pointing out the significant importance of the family in orienting young people towards entrepreneurship in developing countries. Thus, Shittu and Dosunmu [27] demonstrated the importance of the family as a source of role models for young graduates, in a study conducted in Nigeria. Similarly, Denanyoh et al. [28] concluded that the support provided by the university, but especially the moral support provided by the family are among the factors that positively influence entrepreneurship. Other papers [29,30] have also highlighted the key role that family support and entrepreneurship education play in determining students’ entrepreneurial intentions. Additionally, in a study conducted in Romania, Georgescu and Herman [31] observed that students who come from an entrepreneurial family benefit from an informal education and implicitly show a higher entrepreneurial intention than other students. The authors conclude that the exposure to an entrepreneurial context in the family determines a stronger intention to form businesses, but the role of formal education is also important and is a great motivating factor, especially for students who do not have entrepreneurs in their families. 

#### 2.1.2. The Financial Support Expected from Family

Other authors [32] have focused on two types of family support: emotional support, as we presented in the section above, and financial support, both of which are considered equally important in determining entrepreneurial intention. Edelman et al. [33] have considered that financial capital is the very point from which any project is developed, while Bates [34] considered that financial capital alone would not be enough, but it must be associated with human or social capital, in order to ensure the support of entrepreneurship. Rodriguez et al. [35] have shown that the availability of financial capital from the family is the starting point for the development of an entrepreneurial project and its existence positively influences the desire for entrepreneurship. Steier [36] concluded that family funding is probably the most important source of financial resources for young entrepreneurs.

Another research direction in the literature focuses on the additional advantage of family involvement, namely, facilitating access to funding from external sources [37] and consequently, determining the intention of entrepreneurship. Thus, from a financial point of view, family members may appear to provide the entrepreneur with the financial capital needed to start a business or may contribute to obtaining external sources of financing to which the young entrepreneur does not have access [38,39]. In a study on entrepreneurship among young people in Hong Kong and China, Au and Kwan [40] found that youngsters asked for initial family funding only if it brought them lower transaction costs and lower levels of family involvement in the business.

#### 2.1.3. The Individuals’ Independence Regarding the Desire to Start a Business on Their Own

In a study conducted in 50 countries, Sieger et al. [41] identified that from a percentage of 8.8% of students (out of a total of 122,000) who intended to start their own business, only 2.7% wanted to be part of the family business. A similar situation was observed in the case of those who had a period of 5 years of employment. Thus, out of the 38.2% of youngsters that intended to set up a business, only 4.8% were considering a family business.

In addition, Zellweger et al. [21] found that although they are confident in their knowledge and skills, young people do not want to take over the family business because it would mean not being independent. Several research papers [42,43] have found that involvement in a family business creates more stress in the family than other types of employment, which makes young people skeptical about involvement.

A study examining young people’s reasons for starting a business in Nepal [44] showed that profit potential, difficulties in finding jobs, but also the desire to do something new through entrepreneurship are the main factors that motivate young people; only a few of them were motivated by the desire to continue the family business. In another study, analyzing both the expectations of young people and the experience gained by young entrepreneurs, Le Nguyen Doan [45] found that the most important reasons for starting a business were the continuation of family tradition, the need to be independent and the desire to succeed on their own.

#### 2.1.4. The Intention to Form Business Partnerships with Family Members

When making the decision to take part in a family business, young people take into account the fact that the family business brings both advantages and disadvantages. Thus, being part of the same family, they share common values, which can ensure a competitive advantage. At the same time, starting a family business requires a strong commitment and a greater understanding on the part of the family, regarding the flexibility of the work schedule, ensuring greater loyalty and stability. There are also benefits linked to low costs, as family members are willing to accept financial sacrifices for the good of the business. However, there are also a number of disadvantages associated with a family business, such as the lack of experience, which can have an influence on success, and the possibility that family conflicts will transfer to the business.

Thus, studies conducted on the intention of young people to pursue a career in the family business showed that parents’ behavior influences decision-making [46], as well as young peoples’ perception of their parents’ involvement [47]. Other papers that examine trends in family business [38,48] have suggested that the desire for involvement and success depends on family support and communication about the new business. Other studies found that the desire to take over the family business was influenced by several variables such as the age of the business, the way it is run, its complexity, its financial performance and the time required to take over the business [49].

### 2.2. Hypotheses Development

Taking into account the importance of family businesses for the economy of a developing country [50,51,52], we formulated the following hypotheses that could help researchers and other professionals to better understand the driving forces for stimulating intergenerational family businesses. All hypotheses focus on the connections between the variables used in our model: the closeness to family members; the financial support expected from family; the individuals’ independence regarding the desire to start a business on their own; the intention to form business partnerships with family members.

**Hypothesis** **1** **(H1).**
*Closeness to family members has a direct and positive impact on the financial support expected from the family.*


Through this hypothesis, we want to see if close family relations make individuals expect financial support from their family or not, in which case they might want to be financially independent. 

**Hypothesis** **2** **(H2).**
*Closeness to family negatively influences the individuals’ independence regarding the desire to start a business on their own.*


This hypothesis derives from the first hypothesis. Thus, we want to see if individuals who are closer to their families are more prone to starting businesses on their own or with their family members. 

**Hypothesis** **3** **(H3).**
*Closeness to the family has a direct and positive impact on the intention to form business partnerships with family members.*


This hypothesis focuses on the role played by closeness to the family on the intention to involve family members in a potential business developed by an individual. 

**Hypothesis** **4** **(H4).**
*The individuals’ independence regarding the desire to start a business on their own negatively influences the intention to form business partnerships with family members.*


Through this hypothesis, we want to see if independent individuals who want to start a business on their own would still want to involve in one form or another, their family members. 

**Hypothesis** **5** **(H5).**
*The financial support expected from the family has a direct and positive impact on the intention to form business partnerships with family members.*


This hypothesis focuses on the connection between the financial support received by an individual from his/her family and the desire to start a family business because of this reason. 

## 3. Research Methodology

We applied the method of partial least squares structural equation modeling (PLS-SEM), using SmartPLS v.3 software [53]. Our research model is presented in Figure 1 and includes four variables with specific items. Thus, as we can see in Table 1, the closeness to family members has four items; the financial support expected from family has two items; the individuals’ independence regarding the desire to start a business on their own has one item; the intention to form business partnerships with family members has 11 items.

The constructs, their items and the codification used in the model are presented in Table 1. 

The data were collected using a survey, the items in Table 1 were measured using a Likert scale, ranked from total disagreement (1) to total agreement (5). The survey, focusing on the intention of individuals to start a business on their own taking into account their families, was sent by e-mail and social media to 950 individuals from Romania. Out of these, 200 questionnaires were valid. The answers were collected between June and September 2021. The respondents are between 19 and 55 years old, with an average age of 29 years. Because the answers were diverse, we grouped respondents into two groups: those below 29 years old and those above 29 years old. The descriptive statistics and the frequencies are presented in Table 2. Our research focused mostly on younger people because they are the ones who can start an intergenerational family business with both the first generation (their grandparents) and the second generation (their parents). In addition, they exhibited a higher interest in responding to a survey that had a business topic. The respondents are from Romania, a developing country, where the net minimum wage in 2022 is 1,524.00 lei (approx. 310 euros) and almost a third of active adults gain the net wage [54]. Taking into consideration the low income of most employees in Romania and the desire to start a business, we considered it appropriate to choose Romania, a developing country, for our research.

## 4. Results

In order to ensure the convergent validity of the model in Figure 1, we determined the outer loadings of the items for each construct, these being presented in Table 3. The items with the outer loadings below 0.7 were removed from the model. Thus, CLOSE 1 will be removed, as part of the CLOSE construct. Seven items will be removed (PRTN1-3, PRTN7, PRTN 9-11), as part of the PRTN construct. All VIF values are under 5, so the collinearity statistics of the model are assured.

By removing the items with the outer loadings below 0.7, the model changes, which is reflected in Figure 2. We can notice that the strongest effect is from CLOSE to PRTN (0.374), followed by FINS to PRTN (0.352), CLOSE to INDEP (−0.338) and CLOSE to FINS (0.329). The path coefficients with a minus sign show the negative correlation between the variables (CLOSE to INDEP and INDEP to PRTN). CLOSE, FINS and INDEP explain 46.8% of the variance of PRTN, while CLOSE and PRTN explain 11.4% of the variance of INDEP and CLOSE explains 10.8% of the variance of FINS. 

For measuring the construct reliability and validity, we determined Cronbach’s Alpha and Average Variance Extracted (AVE). The results are presented in Table 4. Cronbach’s Alpha values are above 0.6, a value considered acceptable for the model’s reliability [56,57,58]. All values of AVE for all constructs are above 0.6 (high convergent validity) and all values of Composite Reliability are over 0.8 (high internal composite reliability), which shows that the variables in the model are significant.

For the discriminant validity of the model, we used the Fornel–Larcker Criterion. The square roots of AVE are presented in the main diagonal of Table 5. All of them are higher than the correlation values calculated for each construct in relation to the other constructs in the model (in each column), thus ensuring the discriminant validity of the model and its constructs.

To test the model’s significance, we also used Bootstrapping and determined t statistics, *p* values and the confidence interval, as we can see in Table 6. All path coefficients in the model are statistically significant for a 5% significance level, because t statistics are above 1.96 and *p* values are below 0.05.

According to data in Table 6, we can notice that all five hypotheses are validated because zero is not in the confidence interval for either of the hypotheses. The results are summarized in Table 7.

The structural model analysis is done with the blindfolding test which calculates the Q2 values [59,60]. The values of Q2 are above 0 as we can see in Table 8, which shows the predictive relevance of the model. PRTN has the highest value (0.303), followed by INDEP (0.102) and FINS (0.074).

The descriptive statistics of the items retained in the path model are presented in Table 9. The mean higher than 2.5 and the outer loadings above 0.7 also show the importance and the relevance of the items retained in the research model. 

## 5. Discussion

All five hypotheses of our research are validated showing that factors such as the closeness to family members and the financial support received from the family influence the intention to develop partnerships with family members. At the same time, individuals who exhibit more independent behavior are more prone to starting their own businesses without having their families involved. 

Our research confirms hypothesis 1 which shows that there is a direct and positive relationship between closeness to family members and the financial support expected from relatives. This positive relationship creates the premises for founding family businesses between different generations of a family. The role of financial support in the development of family businesses is highlighted by other authors too [61]. 

Hypothesis 2 is validated, with the negative correlation being reflected by the negative value in Figure 2 (path coefficients from CLOSE to INDEP), Table 5 (Fornell–Larcker Criterion) and 6 (Bootstrap results). There is a direct correlation between the two variables but this one is negative. Thus, individuals that are closer to their family members are not so eager to found independent businesses, without taking into account their relatives. Grote [62] analyzes the impact of conflict between generations (so the level of closeness in the family) which puts a strain on the family business. The conflicts between generations can lead to a desire from the younger generations to want to be more independent and start their own business. 

Our statistical findings validated hypothesis 3 which shows a direct relationship between closeness to family members to the intention to start a family business and have family members as partners. This result is in accordance with similar studies in the literature that analyzed the connection between closeness (also named proximity, intimacy, commitment, common beliefs and goals) and the intention to develop or continue a family business [48,63,64,65]. 

The fourth hypothesis is validated and the negative correlation is reflected by the negative value in Figure 2 (path coefficients from INDEP to PRTN), Table 5 (Fornell–Larcker Criterion) and 6 (Bootstrap results). Individuals who show a more independent attitude are not interested in starting a family business or continuing one that exists. Other studies mention the conflict between generations [63] or personal factors such as the desire to be independent or the rebellion of younger generations [66] that affect family businesses. 

Our results show a direct relationship between the financial support expected and received from family members and the intention to develop or continue the family business, thus validating hypothesis 5. Thus, the financial resources remain in the family. There are studies that emphasize the risk that financial support might affect entrepreneurial intentions [33,67] and other studies which recognize the importance of family financing for starting a business [68,69]. Considering the statistics in Romania and the low income gained by most employees [54], the financial support of the family is a start for many youngsters that do not have the possibility to borrow from the bank.

## 6. Conclusions

The main findings of the present research reveal the impact of closeness to family members on the intention to develop intergenerational family businesses (H3). Additionally, the close connection with family members influences the expectancy regarding the financial support received from relatives (H1) which also impacts the decision to initiate business partnerships with the family (H5). Individuals who are not close to their family members show an increased desire to form independent businesses without any connection to their families (H2 and H4). 

### 6.1. Theoretical and Practical Implications

Polin and Golla [70] conducted an analysis regarding entrepreneurship in both developed and developing countries, concluding that the interest in entrepreneurship is higher in developing economies. The findings of our research are useful for the researchers because they can represent a starting point for analyses in other countries with different levels of economic and social development.

From a practical point of view, our research can help political decision-makers to better understand the importance of family for the economic development of a country and create educational and economical strategies that stimulate the creation of family businesses. The role of family businesses in the development of an economy is highlighted in many studies [71,72,73,74]. Such strategies should promote values like family stability, closeness between parents, grandparents and children and the support of families in raising and educating their children. These would create a foundation for close relationships between family members, an increased interest in starting family businesses with the former generations and also ensure the family’s prosperity by at least offering the opportunity for additional income. Meghisan–Toma et al. [75] also confirmed a “significant positive impact of social influence on behavioral intention”. 

### 6.2. Limitations and Future Research Directions

The present research is based on a survey using questionnaires that were distributed online taking into account the restrictions imposed during the COVID-19 pandemic in 2020 and 2021. Thus, our main respondents were individuals with Internet access. Still, there are people in rural areas with a poor Internet connection or even without one who start family businesses because this is the only way to gain an income. Mostly, these family businesses are in the agricultural domain. According to INS [76], 46.2% of Romanians live in rural areas. Additionally, more than 90% of Romanian farms are small ones, oriented toward satisfying the needs of the family and obtaining a small profit by selling the surplus [77]. Less than 10% of the farmers are under 40 years old [78]. These statistics might be explained by the differences between generations, their needs, vision, motivations and objectives. 

Regarding future research directions, more variables can be added, together with items that can affect the intention to start an intergenerational family business, such as the possibility to influence decisions and impact the future of the business [79], the education received in the family regarding the importance of family business [80], the level of their studies, the economic status of the family during one’s childhood and gender. Two-thirds of Romanian entrepreneurs are men [81] and 68% of the female entrepreneurs are working in agriculture [82]. Other factors that could be analyzed are the impact of the pandemic restrictions on youngsters enrolled in economic faculties [83] and the role played by motivation at work in the decision to become an entrepreneur [84]. In addition, the sample structure could be diversified in order to include people who live in rural areas and individuals who do not have an internet connection, so using a face-to-face questionnaire would be more appropriate for this population. Other research directions can focus on the extension of the research to other countries, with different levels of economic development in order to make a comparative analysis.

## Figures and Tables

**Figure 1 behavsci-12-00181-f001:**
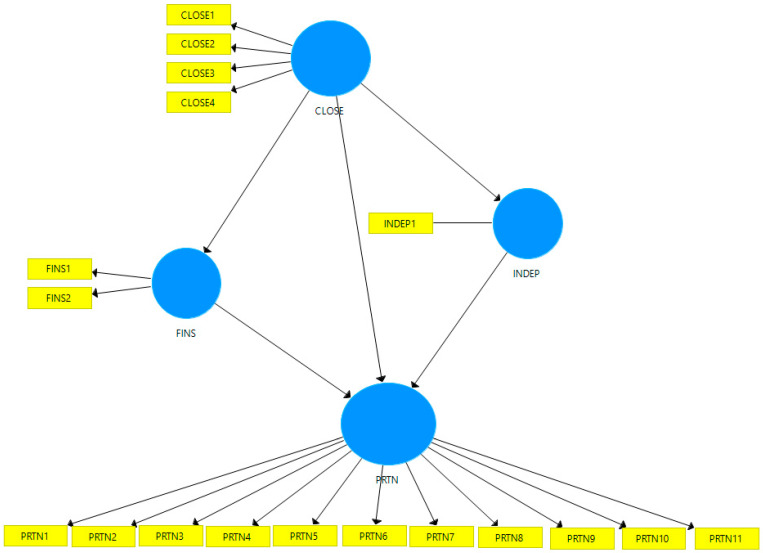
The path model of the research. Source: Model created by authors with SmartPLS v.3 software.

**Figure 2 behavsci-12-00181-f002:**
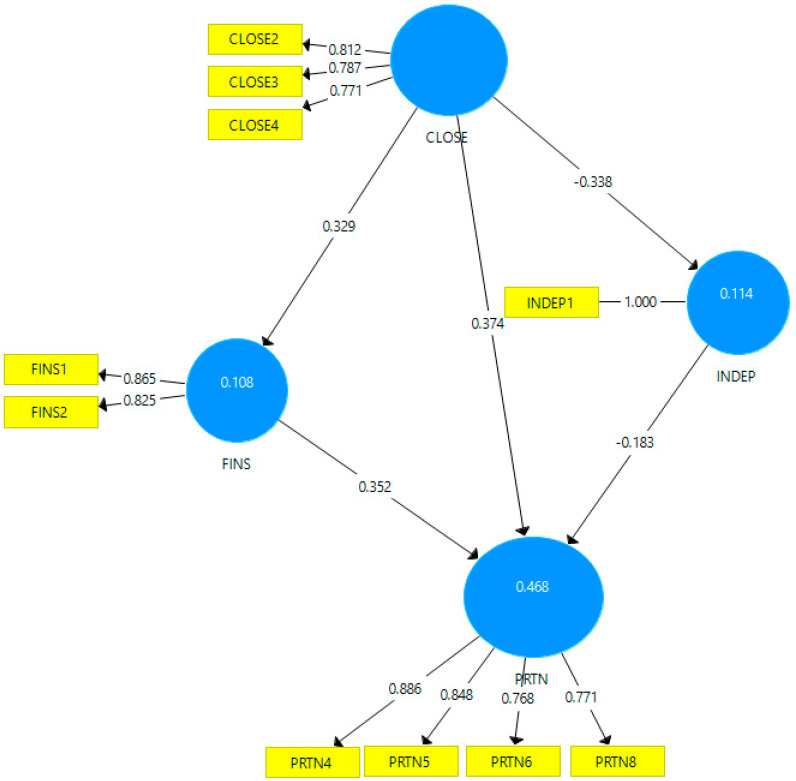
Path coefficients and outer loadings of the model. Source: Developed with SmartPLS v.3 software.

**Table 1 behavsci-12-00181-t001:** Description of the model.

Constructs (Code)	Items	Item Codes
Closeness to family members (CLOSE)	My family is near me and supports me by all means.	CLOSE1
	When I make decisions, I ask for my family’s advice.	CLOSE2
	When I have problems, I share them with my family.	CLOSE3
	In order to start something, I need the support from my family.	CLOSE4
The financial support expected from family (FINS)	I can convince my family to offer me financial support to start a business on their own.	FINS1
	If my family members are suppliers for my business, they support me financially.	FINS2
The individuals’ independence regarding the desire to start a business on their own (INDEP)	I would like to have a completely independent business, without any interaction with my family.	INDEP1
The intention to form business partnerships with family members (PRTN)	I would feel safer if my family members would be suppliers for my business.	PRTN1
	I would include my family members in my plans to start a business on their own.	PRTN2
	Having good business relations with my family would reduce the stress for me.	PRTN3
	I would prefer to have an intergenerational family business with my relatives than a completely independent business.	PRTN4
	I would prefer to have a family business with my parents.	PRTN5
	I would prefer to have a joint business with my relatives.	PRTN6
	I would involve my parents in my business as employees.	PRTN7
	I would involve my parents in my business as subcontractors.	PRTN8
	If I started a business, I would use the competencies of my family members.	PRTN9
	I would develop a business with my grandparents (the first generation).	PRTN10
	I would develop a business with my parents (the second generation).	PRTN11

Source: The survey is created as part of the INTERGEN-2 consortium (Acknowledgements).

**Table 2 behavsci-12-00181-t002:** Descriptive statistics and frequency for the respondents’ age.

Age	Frequency	Percent	Mean	Minimum	Maximum
19–29 years old	123	61.5	29.625	19	55
30–55 years old	77	38.5			

Source: Own analysis using JASP software [55].

**Table 3 behavsci-12-00181-t003:** Items’ outer loadings and outer VIF values.

Item Codes	Outer Loadings	Outer VIF Values
CLOSE1	0.612	1.202
CLOSE2	0.811	1.606
CLOSE3	0.716	1.446
CLOSE4	0.723	1.289
FINS1	0.845	1.227
FINS2	0.846	1.227
INDEP1	1.000	1.000
PRTN1	0.632	1.760
PRTN2	0.674	2.292
PRTN3	0.679	1.940
PRTN4	0.828	3.061
PRTN5	0.834	3.361
PRTN6	0.735	2.433
PRTN7	0.619	1.585
PRTN8	0.735	2.029
PRTN9	0.631	1.686
PRTN10	0.489	1.513
PRTN11	0.695	2.121

Source: Own analysis using SmartPLS v.3.

**Table 4 behavsci-12-00181-t004:** Construct reliability and validity.

Construct	Cronbach’s Alpha	rho_A	Composite Reliability	AVE
CLOSE	0.699	0.702	0.833	0.624
FINS	0.601	0.607	0.833	0.714
INDEP	1.000	1.000	1.000	1.000
PRTN	0.836	0.843	0.891	0.671

Source: Own analysis using SmartPLS v.3.

**Table 5 behavsci-12-00181-t005:** Fornell–Larcker Criterion.

Construct	CLOSE	FINS	INDEP	PRTN
CLOSE	0.790			
FINS	0.329	0.845		
INDEP	−0.338	−0.300	1.000	
PRTN	0.552	0.530	−0.415	0.819

Source: Developed with SmartPLS v.3 software.

**Table 6 behavsci-12-00181-t006:** Bootstrapping results.

	T Statistics	*p*–Values	Confidence Interval Bias Corrected
CLOSE- > FINS	5.017	0.000	(0.186, 0.447)
CLOSE- > INDEP	4.983	0.000	(−0.469, −0.201)
CLOSE- > PRTN	6.191	0.000	(0.228, 0.475)
INDEP- > PRTN	2.704	0.007	(−0.304, −0.054)
FINS- > PRTN	5.589	0.000	(0.244, 0.480)

Source: Developed with SmartPLS v.3 software.

**Table 7 behavsci-12-00181-t007:** Hypotheses validation.

Hypothesis	Results
CLOSE- > FINS (H1)	Accepted
CLOSE- > INDEP (H2)	Accepted
CLOSE- > PRTN (H3)	Accepted
INDEP- > PRTN (H4)	Accepted
FINS- > PRTN (H5)	Accepted

Source: Own analysis.

**Table 8 behavsci-12-00181-t008:** Blindfolding test.

Construct	SSO	SSE	Q^2^ (=1-SSE/SSO)
CLOSE	600.000	600.000	
FINS	400.000	370.483	0.074
INDEP	200.000	179.665	0.102
PRTN	800.000	557.987	0.303

Source: Developed with SmartPLS v.3 software.

**Table 9 behavsci-12-00181-t009:** Descriptive statistics of the items in the model.

Item Codes	Mean	Standard Deviation	Outer Loading
CLOSE2	3.675	1.371	0.812
CLOSE3	3.455	1.424	0.787
CLOSE4	2.855	1.433	0.771
FINS1	3.495	1.393	0.865
FINS2	3.895	1.136	0.825
INDEP1	3.335	1.461	1.000
PRTN4	3.085	1.318	0.886
PRTN5	3.360	1.378	0.848
PRTN6	2.545	1.352	0.768
PRTN8	3.160	1.405	0.771

Source: Developed with SmartPLS v.3 and JASP software.

## Data Availability

Not applicable.

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
