# Peer review of "A Multivariate Analysis of the Interest in Starting Family Businesses within a Developing Economy"

_behavsci, 2022, doi:10.3390/bs12060181_

Round 1
Reviewer 1 Report
In the reviewer's opinion, the article meets the requirements, nevertheless, a few points should be pointed out:
- the study focuses on one country, a comparative analysis would be more interesting, but at this moment it is not possible to carry it out and describe in the paper
- the literature review should refer to similar studies in Central and Eastern Europe - if possible
- it is worth emphasizing the importance and specificity of family relations in Romania (also in comparison with other countries)
- I would recomment a short commentary on the hypotheses - point 2.2.
- we should be aware that the number of questionnaires on which the analysis is based could be greater
- I would recommend more extensive comments on Tables 4-9
Despite some comments, I find the analysis interesting and correct
Author Response
Dear Reviewer,
Thank you for your observations and for the opportunity to improve our manuscript!
We are very grateful for taking the time to analyze the paper and make very useful, encouraging and thoughtful comments and recommendations.
We have read the evaluation carefully and, based on the review reports, we performed significant revisions of our manuscript, as requested, highlighted with red into the manuscript, respectively:
Response to Reviewer 1 Comments
Point 1: The study focuses on one country, a comparative analysis would be more interesting, but at this moment it is not possible to carry it out and describe in the paper
Response 1: Thank you for your correct observation. Even at this point is not possible to that, I emphasised the idea in conclusions, in the part where we describe future research directions, because the idea is a very good one for extending our research.
Point 2: The literature review should refer to similar studies in Central and Eastern Europe - if possible.
Response 2: We added in the literature review two studies from Bulgaria and Poland (references 15 and 16).
Point 3: It is worth emphasizing the importance and specificity of family relations in Romania (also in comparison with other countries).
Response 3: In the literature review section, we added a paragraph analysing the importance of family relation in countries like Romania and others in Central and Eastern Europe emphasizing the connection made in the literature with the communist past of these countries.
Point 4: I would recommend a short commentary on the hypotheses - point 2.2.
Response 4: We added a short explanatory phrase for each of the hypotheses.
Point 5: We should be aware that the number of questionnaires on which the analysis is based could be greater
Response 5: In the research methodology section, we described the way we collected data. We sent the survey to 950 individuals and we got only 200 valid questionnaires. As you correctly mention, the sample could have been higher. Thus, we decided to use the method of partial least squares structural equation modelling (PLS-SEM) which requires a minimum of 150 responses to be considered accepted from a scientifical and statistical point of view. Still, we mentioned in the conclusion section a few of the limitations, among which there is also the COVID-10 pandemic and the restriction in approaching individuals face to face, thus affecting the number of online responses we got.
Point 6: I would recommend more extensive comments on Tables 4-9
Response 6: We want to thank you for your recommendation because, indeed, we missed a brief explanation for table 9 which was now added.
Values in tables 4 to 9 were calculated with Smart-PLS software which was mentioned in the research methodology section. All indicators in these tables are calculated with the software and the values were briefly explained to show the statistical relevance of the model we used.
More explanations for the indicators would not bring more value to the paper and would create more confusion to the usual reader because they are very specific to statistics, mathematics and not related to the topic of our research.
PLS method has specific steps to follow and it is difficult to add an extensive explanation of the method and all of the indicators in our paper.
Still, we included a few references that explain the method per se and the relevance of a value or another for the indicators we calculated with the Smart-PLS software.
- Ringle, C.M.; Wende, S.; Becker, J.M. SmartPLS 3. Boenningstedt: SmartPLS GmbH, 2015. http://www.smartpls.com.
- JASP Team, JASP, Version 0.16, computer software, 2021.
- Creswell, J.W. Educational research - planning, conducting, and evaluating quantitative and qualitative research, 4th Ed. Pearson Merril Prentice Hall: New Jersey, 2010.
- Van Griethuijsen, R.A.; van Eijck, M.W.; Haste, H.; den Brok, P.J.; Skinner, N.C.; Mansour, N.; Gencer, A.S.; BouJaoude, S. Global Patterns in Students’ Views of Science and Interest in Science. Research in Science Education 2015, 45, pp. 581–603. https://doi.org/10.1007/s11165-014-9438-6.
- Taber, K.S. The Use of Cronbach’s Alpha When Developing and Reporting Research Instruments in Science Education. Research in Science Education 2018, Vol. 48, pp. 1273–1296. https://doi.org/10.1007/s11165-016-9602-2.
- Geisser, S. A Predictive Approach to the Random Effect Model. Biometrika 1974, Vol. 61, No. 1, pp. 101–107. https://doi.org/10.2307/2334290.
- Stone, M. An Asymptotic Equivalence of Choice of Model by Cross-Validation and Akaike’s Criterion. Journal of the Royal Statistical Society, Series B (Methodological) 1977, Vol. 39, No. 1, pp. 44–47. http://www.jstor.org/stable/2984877.
Reviewer 2 Report
This is a nice paper. I would love to recommend some revisions:
1. Page 2 line 45-51 is a paragraph; however, it is not perfectly constructed, as it still needs 2 more sentences to stand a better paragraph.
2. I still do not see sufficient discussions of previous researches in your introduction that may warrant the potential research niches as your paper present. You may add it between paragraph 2 and 3 to explicitly point your paper specific contribution, comparable to the past studies.
3. Line 82-85 is just a sentence. Please mind the difference between a sentence and a paragraph.
4. In the conceptual background, You used so many transitional conjuntion like however, nevertheless, etc as an opening of a paragraph. It is not advisable as their function is best to bridge one sentence idea with another. You clearly need a better proofreading.
5. Line 169-171 is a sentence, not a paragraph.
6. The formulation for hypothesis 1 is still vague. Do you not have direct references that closeness to family member would secure their financial support. You need to provide strong citation regarding this formulation, and if not possible, you may construct the logical presentation regarding the hypothesis formulation. This is applicable to all of hypotheses.
7. The indicator for indep1 is a negative sentence. It is in the contrary to the financial support variable. reading your paper's result has to be conducted carefully.
8. The table arrangement has to be conducted systemically. table 9 has to support the arguments in line 250 and so forth. Thus, you must relocate table 9 after the line 250-256. The tables are to wide and can be arranged in to lesser table. You may check how Hair et al. (2014) provide a recommendation for the table construction to make an easier presentation.
9. The discussion is still premature. You must avoid any statistical numberings in this section, as we (readers) can obtain the information from the result section. You need to explain whether your finding is significant/not significant; how previous paper discusses these issues; how your paper's findings contribute to the previous discussions; and what implications it may provide.
Author Response
Dear Reviewer,
Thank you for your observations and for the opportunity to improve our manuscript!
We are very grateful for taking the time to analyze the paper and make very useful, encouraging and thoughtful comments and recommendations.
We have read the evaluation carefully and, based on the review reports, we performed significant revisions of our manuscript, as requested, highlighted with red into the manuscript, respectively:
Response to Reviewer 2 Comments
Point 1: Page 2 line 45-51 is a paragraph; however, it is not perfectly constructed, as it still needs 2 more sentences to stand a better paragraph.
Response 1: We re-read the sentences in the introduction and we grouped them in two main paragraphs. In the previous version, we had a very short paragraph indeed. Thank you for your observation.
Point 2. I still do not see sufficient discussions of previous researches in your introduction that may warrant the potential research niches as your paper present. You may add it between paragraph 2 and 3 to explicitly point your paper specific contribution, comparable to the past studies.
Response 2. We added more references at the beginning of the literature review section to present similar studies focused on family businesses and the importance of family relations in Central and Eastern European countries.
The paragraph added is the following:
“Other studies also analyzed the factors that impact the development of family businesses in Central and Eastern Europe [15,16]. Thus, Demirova and Ahmedova [15] (p.18) high-light the role played by “shared values and beliefs”, “shared goals”, “family spirit”, financial independence, “flexibility”, the specific organizational culture for the success of Bulgarian family businesses. Marjański and Sułkowski [16] conducted research on Polish family businesses emphasizing their importance for the country’s economy and also the fact that these small entities are sometimes better in satisfying the needs of niche markets than larger companies. The analysis of family businesses in Central and Eastern European countries should be seen through the lens of family relations and their role for these countries which had a similar past marked by communism. Robila [17] mention the importance of family ties in countries like Romania or Hungary where youngsters decide to remain in the same household with their parents because of their limited finances. This is an important motivation for uniting efforts to develop a family business [16]. Also, Robila [17] explain the strength of family relations in former communist countries like Romania because of those times in which you could trust only your close family”.
The references used are:
Demirova S.; Ahmedova S. Analytical study of family business in Bulgaria. Innovations 2020, Vol. 8, No. 1, pp. 18-20.
Marjański A.; Sułkowski, Ł. The evolution of family entrepreneurship in Poland: main findings based on surveys and interviews from 2009-2018. Entrepreneurial Business and Economics Review 2019, Vol. 7, No. 1, pp. 95-116. https://doi.org/10.15678/EBER.2019.070106
Robila, M. Families in Eastern Europe: Context, Trends and Variations. In Families in Eastern Europe (Contemporary Perspectives in Family Research, Vol. 5), Robila, M. (Ed.). Emerald Group Publishing Limited, Bingley, 2004; pp. 1-14. https://doi.org/10.1016/S1530-3535(04)05001-0
Point 3. Line 82-85 is just a sentence. Please mind the difference between a sentence and a paragraph.
Response 3. Thank you for your observation. We moved the sentence in the previous paragraph because its meaning was connected to it.
Point 4. In the conceptual background, you used so many transitional conjunctions like however, nevertheless, etc as an opening of a paragraph. It is not advisable as their function is best to bridge one sentence idea with another. You clearly need a better proofreading.
Response 4. We carefully checked the text, eliminated redundant words and also integrated sentences in the appropriate paragraphs.
Point 5. Line 169-171 is a sentence, not a paragraph.
Response 5. We corrected this construction and also others we identified within the text. Thank you for bringing them into our attention.
Point 6. The formulation for hypothesis 1 is still vague. Do you not have direct references that closeness to family member would secure their financial support. You need to provide strong citation regarding this formulation, and if not possible, you may construct the logical presentation regarding the hypothesis formulation. This is applicable to all of hypotheses.
Response 6. We added a short explanation of each hypothesis we introduced. The variables in the five hypotheses are the variables used in our model so we tried to maintain the name of the variable used in the model. Thus, as you correctly noticed, we understand that this was not clearly stated so we added a paragraph above the hypotheses to explain the logic behind the way we formulated each of them.
Point 7. The indicator for indep1 is a negative sentence. It is in the contrary to the financial support variable. reading your paper's result has to be conducted carefully.
Response 7. Indep1 refers to the question included in the survey which was: I would like to have a completely independent business, without any interaction with my family. (Table 1).
The value for Fornell-Larcker Criterion (Table 5) for INDEP in relation to financial support (FINS) is indeed negative. We did not include in the model a hypothesis that connects these 2 variables but the PLS method using Smart PLS software analyses the bilateral connections from all variables in the model to test the statistical significance.
The bootstrap test in table 6 explains the relationship between the variables in the 5 hypotheses we formulated at the beginning of our research.
Point 8. The table arrangement has to be conducted systemically. table 9 has to support the arguments in line 250 and so forth. Thus, you must relocate table 9 after the line 250-256. The tables are too wide and can be arranged in to lesser table. You may check how Hair et al. (2014) provide a recommendation for the table construction to make an easier presentation.
Response 8. We dimensioned all tables to fit the contents. Thank you for the reference provided. We also added a short explanation above table 9. Regarding the order of the tables, we tried to comply with the steps in the PLS method which has a specific sequence.
Point 9. The discussion is still premature. You must avoid any statistical numberings in this section, as we (readers) can obtain the information from the result section. You need to explain whether your finding is significant/not significant; how previous paper discusses these issues; how your paper's findings contribute to the previous discussions; and what implications it may provide.
Response 9. We eliminated from the discussion the statistical values for t and p and kept only the findings of our research and the references to other studies that analysed the topic. The theoretical and practical implications of our research are included in the conclusion section, paragraph in which we emphasized the utility of our findings.
Reviewer 3 Report
Very interesting and complex topic that the authors have chosen. I like the paper but I think that there are a few things that should be address:
1) Presentation. There are some things in the presentation in the paper that need to be improved, such as the section header “Results” in page 6 and then all the related text in the next page. It does not look good like that.
2) I think that the paper also would benefit from some comparison with regional peers. No need to do quantitative analysis but a qualitative assessment of the applicability of the analysis to other countries in the region.
3) They authors also make a very interesting point about the limitations (due to internet connectivity…) and the analysis of rural business, which are important in the Romania economy. I think that it would be interesting to expand this section with a bit more details. For example what type of agricultural business are predominant? Are they export oriented?
4) In some other countries, such as for instance China, it is frequent to find generational transfer issues with the new generations not wanting the hard work and sometimes unglamorous work inherited from the family business (which in many cases are very successful). I think that it would be interesting to see if that’s also the case in Romania (again qualitative description would be enough).
5) Another important comments made by the authors that I would suggest to be expanded is regarding the informal education received by individuals from families with entrepreneurs. I think it would be very interesting to expand this section.
6) Another suggestion is to control for gender or at least try to qualitative describe the impact that gender might have on the issue, for example the proportion of female/male entrepreneurs. Are they business in which there is a mark gender difference?
I hope that you find the comments helpful.
Author Response
Dear Reviewer,
Thank you for your observations and for the opportunity to improve our manuscript!
We are very grateful for taking the time to analyze the paper and make very useful, encouraging and thoughtful comments and recommendations.
We have read the evaluation carefully and, based on the review reports, we performed significant revisions of our manuscript, as requested, highlighted with red into the manuscript, respectively:
Response to Reviewer 3 Comments
- Presentation. There are some things in the presentation in the paper that need to be improved, such as the section header “Results” in page 6 and then all the related text in the next page. It does not look good like that.
Response: We corrected the problem and checked the paper for similar issues.
- I think that the paper also would benefit from some comparison with regional peers. No need to do quantitative analysis but a qualitative assessment of the applicability of the analysis to other countries in the region.
Response: As the other reviewers also recommended this, we added a few paragraphs in the literature review to emphasize other studies conducted in Romania, Bulgaria, Poland in relation to family businesses and their role especially in connection with the communist past of many countries in Central and Easter Europe.
- They authors also make a very interesting point about the limitations (due to internet connectivity…) and the analysis of rural business, which are important in the Romania economy. I think that it would be interesting to expand this section with a bit more details. For example what type of agricultural business are predominant? Are they export oriented?
Response: We added some statistical details from the official statistics provided by the Ministry of Agriculture and Rural Development in Romania:
Also, more than 90% of Romanian farms are small ones, oriented for satisfying the needs of the family and obtaining a small profit by selling the surplus [79].
- In some other countries, such as for instance China, it is frequent to find generational transfer issues with the new generations not wanting the hard work and sometimes unglamorous work inherited from the family business (which in many cases are very successful). I think that it would be interesting to see if that’s also the case in Romania (again qualitative description would be enough).
Response: We did not find studies focused especially on generational issues in Romania in relation to family businesses but this might be seen in some problems identified at the level of farm family businesses. Less and less youngsters are interested in agriculture.
Thus, we added in the conclusion part the following paragraph: Less than 10% of the farmers are under 40 years old [80]. These statistics might be explained by the differences between generations, their needs, vision, motivations and objectives.
- Another important comment made by the authors that I would suggest to be expanded is regarding the informal education received by individuals from families with entrepreneurs. I think it would be very interesting to expand this section.
Response: Reference 31 in our paper (Georgescu, M.-A.; Herman, E. The Impact of the Family Background on Students’ Entrepreneurial Intentions: An Empirical Analysis. Sustainability 2020, Vol. 12, No. 11, 4775. https://doi.org/10.3390/su12114775.) analyzed this aspect. We expanded the section by better explaining their findings. Thus, we added the following: The authors conclude that the exposure to an entrepreneurial context in the family determines a stronger intention to form businesses, but the role of formal education is also important and is a great motivating factor especially for the students who do not have entrepreneurs in their families.
As for our research, we did not approach this educational part but we mentioned it in the conclusion, in the paragraph about future research directions because we perceive it as an important factor that might influence entrepreneurial intentions.
- Another suggestion is to control for gender or at least try to qualitative describe the impact that gender might have on the issue, for example the proportion of female/male entrepreneurs. Are they business in which there is a mark gender difference?
Response. We added gender as a variable that should be considered as a future research direction (in the conclusion part) and added two references (83, 84) about the prevalence of men among entrepreneurs (two thirds) in Romania and that 68% of women entrepreneurs are working in agriculture.
Round 2
Reviewer 2 Report
In the discussion section. Do not put H1. ..... comments. They are better integrated in the paragraph. It is better to write them down like "Our statistical finding confirm the hypothesis 1 in the analysis. This finding is in line with previous researches ......
You may add 2-3 references from the behavioral sciences as your citation as it connects your paper to the previously published paper in the journal.
Best,
Author Response
Dear Reviewer,
Thank you for your observations and for the opportunity to improve our manuscript!
We are very grateful for taking the time to analyze the paper and make very useful, encouraging and thoughtful comments and recommendations.
We have read the evaluation carefully and, based on the review reports, we performed significant revisions of our manuscript, as requested, highlighted with red into the manuscript, respectively:
Response to Reviewer 2 Comments
Point 1: In the discussion section. Do not put H1. ..... comments. They are better integrated in the paragraph. It is better to write them down like "Our statistical finding confirm the hypothesis 1 in the analysis. This finding is in line with previous researches ......
Response 1: All hypotheses were integrated in the paragraphs.
Point 2. You may add 2-3 references from the behavioral sciences as your citation as it connects your paper to the previously published paper in the journal.
Response 2. There was added a paragraph in the conclusion section citing 2 papers, 85 and 86.